# Complete Mitochondrial Genome of Scolytoplatypodini Species (Coleoptera: Curculionidae: Scolytinae) and Phylogenetic Implications

**DOI:** 10.3390/genes14010162

**Published:** 2023-01-06

**Authors:** Guangyu Yu, Shengchang Lai, Song Liao, Yufeng Cao, Weijun Li, Chengpeng Long, Hagus Tarno, Jianguo Wang

**Affiliations:** 1Laboratory of Invasion Biology, School of Agricultural Sciences, Jiangxi Agricultural University, Nanchang 340045, China; 2Forest Protection, Forestry College, Nanjing Forest University, Nanjing 210036, China; 3College of Forestry and Biotechnology, Zhejiang Agricultural and Forestry University, Lin’an 311300, China; 4Department of Plant Pests and Diseases, Faculty of Agriculture, Universitas Brawijaya, Jl. Veteran, Malang 65145, Indonesia

**Keywords:** Scolytinae, Scolytoplatypodini, mitochondrial genome, phylogenetic analysis

## Abstract

The complete mitochondrial genomes (mitogenomes) of beetles in the tribe Scolytoplatypodini (genus *Scolytoplatypus*) were sequenced and annotated. These included *Scolytoplatypus raja* (15,324 bp), *Scolytoplatypus sinensis* (15,394 bp), *Scolytoplatypus skyliuae* (15,167 bp), and *Scolytoplatypus wugongshanensis* (15,267 bp). The four mitogenomes contained 37 typical genes, including 13 protein-coding genes (PCGs), 22 transfer RNA genes (tRNAs), and 2 ribosomal RNA genes (rRNAs). The gene orientation and arrangement of the four mitogenomes were similar to other Coleoptera mitogenomes. PCGs mostly started with ATN and terminated with TAA. The Ka/Ks ratio of 13 PCGs in the four species revealed that *cox1* had the slowest evolutionary rate and *atp8* and *nad6* had a higher evolutionary rate. All tRNAs had typical cloverleaf secondary structures, but *trnS1* lacked dihydrouridine arm. Partial tRNAs lost the discriminator nucleotide. The *trnY* did not possess the discriminator nucleotide and also lost three bases, showing a special amino-acyl arm. Bayesian inference (BI) and maximum likelihood (ML) methods were conducted for phylogenetic analyses using 13 PCGs. Scolytoplatypodini was clustered with Hylurgini and Hylastini, and the monophyly of Scolytoplatypodini was supported. The four newly sequenced mitogenomes increase understanding of the evolutionary relationships of Scolytoplatypodini and other Scolytinae species.

## 1. Introduction

The Scolytoplatypodini is a tribe of wood-boring ambrosia beetles in the subfamily Scolytinae (Curculionidae). They farm fungi as the food source for larvae and adults in their gallery systems [1,2]. The females of most species have a unique mycangial structure located on the pronotum for carrying fungal spores [3]. However, this mycangial structure is absent in some species [4]. Except for *Scolytoplatypus bombycinus*, which is host-specific, other Scolytoplatypodini species have no selection propensity for hosts [4]. Scolytoplatypodini species are mostly secondary borers and usually attack small branches of dead trees. They do not usually attack healthy trees, but some species will attack healthy living trees [5]. There are two genera in Scolytoplatypodini: *Scolytoplatypus* Schaufuss 1891 and *Remansus* Jordal 2013, including 60 species. *Scolytoplatypus* occur mainly in the Afrotropical and Oriental regions, with a few occurrences in temperate regions of Japan and India, and *Remansus* are only found in Madagascar [6]. Phylogenetic analysis studies on Scolytoplatypodini have mostly focused on selected gene fragments. Jordal [6] conducted a phylogenetic analysis of the Scolytoplatypodini in Africa and Asia using four gene fragments and found that the Asian species of *Scolytoplatypus* clearly differ from the African species of *Scolytoplatypus*. Additional molecular data are needed to explore the taxonomic status and systematic relationship of *Scolytoplatypus* and Scolytoplatypodini in Scolytinae. Among various animal groups, complete mitochondrial genomes (mitogenomes) phylogenetic analyses were credible [7]. Mitogenomes also have been applied to evaluate population genetics, phylogeography, and systematics at different taxonomic levels [8]. Therefore, the mitogenome can be a powerful tool for determining the relationship of Scolytoplatypodini within the Scolytinae.

DNA sequencing technology has allowed the sequencing of many insect mitogenomes [9,10]. The typical mitogenome of insect has closed-circular and double-stranded DNA, containing 13 protein-coding genes (PCGs), 22 transfer RNA genes (tRNAs), two ribosomal RNA genes (rRNAs), and an A+T-rich region (D-loop), totaling 37 genes [8,11,12]. Mitogenomes are commonly used in phylogenetic analysis of insect lineages because of their maternal inheritance, rare recombination, relatively high evolutionary rate, and conserved gene components [11,13]. The mitogenome was used to reconstruct the phylogeny of weevils (Curculionoidea) and show that Scolytinae forms a separate lineage [7,14]. Representative bark and ambrosial beetles of Scolytinae have also been studied for phylogeny and taxonomy based on mitogenomes [15].

In this study, the mitogenome of *S. raja* (Blandford, 1893); *S. sinensis* (Tsai and Huang, 1965); *S. skyliuae* (Liao, Lai, and Beaver, 2022); and *S. wugongshanensis* (Liao, Lai, and Beaver, 2022) were provided and analyzed. These complete mitogenomes are first reported in the Scolytoplatypodini. We also explored the genome structure, nucleotide compositions, codon usage, gene overlaps, intergenic spacers, tRNA secondary structure, and the D-loop. Reconstructing the phylogenetic tree of Scolytoplatypodini based on 13 PCGs by Bayesian Inference (BI) and Maximum Likelihood (ML) methods.

## 2. Materials and Methods

### 2.1. Sample Collection and DNA Extraction

The samples for genome sequencing of *S. raja*, *S. sinensis*, *S. skyliuae*, and *S. wugongshanensis* were collected in China (Table 1) and identified according to the corresponding key [1]. All of the fresh samples were placed in absolute alcohol, preserved at −20 °C, and stored at the School of Agricultural Sciences, Jiangxi Agricultural University, Nanchang, Jiangxi 340045, China. The total genomic DNA was extracted from the leg muscle tissue of individual specimens using the Universal Genomic DNA Kit (Jiangsu, China), following manufacturer instructions. DNA was stored for sequencing at −20 °C.

### 2.2. Mitogenome Sequencing and Assembly

Following manufacturer recommendations, NEB Next^®^ Ultra™ DNA Library Prep Kit was used to generate the sequencing library for Illumina (Lincoln, NE, USA) and add index code to each sample. Using Illumina PE Cluster Kit (Illumina, Lincoln, NE, USA), we performed the clustering of the index-coded samples on a cBot Cluster Generation System, according to manufacturer instructions. When the cluster was generated, we used the Illumina platform (NovaSeq 6000) to sequence the DNA libraries and generated 150 bp paired-end reads. After removing low-quality sequences, the reads were assembled into a complete mitogenome by MitoFlex [16].

### 2.3. Sequence Annotation and Analyses

MITOS [17] web server was used to determine 37 genes and all tRNA secondary structures under default parameters. According to the MITOS predictions, secondary structures for tRNAs were manually drawn with Microsoft PowerPoint 2017. The remaining PCGs and rRNA were manually corrected in Geneious 8.1.3 [18]. To draw the mitogenome circular map, we used OrganellarGenomeDRAW (OGDRAW) version 1.3.1 [19]. The nucleotide composition and relative synonymous codon usage (RSCU) were calculated by MEGA X [20] and PhyloSuite v 1.2.2 [21], respectively. Strand asymmetry was calculated by the formulas: GC-skew = [G − C]/[G + C] and AT-skew = [A − T]/[A + T] [22]. DnaSP v6.12.03 was used to calculate the nucleotide diversity (Pi) and nonsynonymous (Ka)/synonymous (Ks) mutation rate ratios of 13 PCGs [23].

### 2.4. Phylogenetic Analysis

A total of 34 mitogenomes (Table 2) were used to construct the phylogenetic tree. PhyloSuite v 1.2.2 [21] was used to download the data onto mitogenomes from GenBank (except *S. raja*, *S. skyliuae*, *S. sinensis*, and *S. wugongshanensis*). All sequences were standardized and extracted 13 PCGs by PhyloSuite v 1.2.2 [21]. The 13 PCGs (excluding the stop codons) of the 34 beetle species were aligned individually using codon-based multiple alignments with MAFFT v7.313 software with default settings [24]. Gblocks v 0.91b software was used to remove the intergenic gaps and ambiguous sites [25], and all PCGs genes were concatenated in PhyloSuite v 1.2.2 [21]. The best partitioning scheme and evolutionary models for constructing BI and ML trees were selected by PartitionFinder2 [26], with a greedy algorithm, BIC criterion, and the gene and codon model. The results are presented in Appendix A.

MrBayes v 3.2.6 [31] and IQ-TREE v.1.6.8 [32] software were employed in PhyloSuite v 1.2.2 to construct the BI and ML phylogenetic trees [21] (refer to Du et al. [15]). *Sitophilus oryzae* and *Sitophilus zeamais* were used as the out-group. ML phylogeny was inferred using IQ-TREE v.1.6.8 [32] under the Edge-linked partition model for 10,000 ultrafast bootstraps [33]. BI phylogeny was inferred using MrBayes 3.2.6 [31] under the partition model (two parallel runs, 2,000,000 generations), in which the initial 25% of sampled data were discarded as burn-in. A PSRF close to 1.0 and a standard deviation of split frequencies below 0.01 were accepted.

## 3. Results and Discussion

### 3.1. Mitogenome Organization and Nucleotide Composition

The complete mitogenome lengths of *S. raja*, *S. sinensis*, *S. skyliuae*, and *S. wugongshanensis* were 15,324 bp, 15,394 bp, 15,166 bp, and 15,267 bp, respectively. These mitogenomes have a similar structure; all mitogenomes exhibited the typical insect mitogenome structure, closed-circular and double-stranded DNA, containing 13 PCGs, 22 tRNAs, 2 rRNAs, and a D-loop (Figure 1). There were 23 genes encoded by the majority strand (J-strand), including 9 PCGs and 14 tRNAs. The remaining 14 genes were encoded by the minority strand (N-strand), including 4 PCGs, 8 tRNAs, and 2 rRNAs (Appendix A).

The basic composition of *S. raja* was A = 39.4%, T = 36.4%, C = 14.8%, and G = 9.4; *S. sinensis* was A = 39.5%, T = 36.3%, C = 15.0%, and G = 9.2%; *S. skyliuae* was A = 39.6%, T = 36.7%, C = 15.5%, and G = 8.2%; *S. wugongshanensis* was A = 39.0%, T = 37.2%, C = 15.6%, and G = 8.2% (Appendix A). In all four species, the nucleotide composition of the whole mitogenome exhibited a distinct A/T bias: 75.8% (*S. raja*), 75.8% (*S. sinensis*), 76.3% (*S. skyliuae*), and 76.2% (*S. wugongshanensis*). A higher A/T bias was also found in *Scolytinae* mitogenomes [15]. The AT-skew ranged from 0.02 (*S. wugongshanensis*) to 0.04 (*S. raja*, *S. sinensis*, and *S. skyliuae*), the GC-skew ranged from −0.31 (*S. wugongshanensis*) to −0.22 (*S. raja*) (Appendix A).

### 3.2. Protein-Coding Genes

The total lengths of the 13 PCGs of *S. raja*, *S. sinensis*, *S, skyliuae*, and *S. wugongshanensis* were 11,070 bp, 11,062 bp, 11,023 bp, and 11,025 bp, respectively (Appendix A). Of these 13 PCGs, 9 PCGs are located at the J-strand, the other 4 PCGs were encoded by the N-strand (Figure 1, Appendix A). The whole 13 PCGs AT-skew and GC-skew were all negative; the AT-skew were −0.13 (*S. raja* and *S. skyliuae*) and −0.14 (*S. sinensis* and *S. wugongshanensis*), the GC-skew were −0.04 (*S. raja* and *S. sinensis*) and −0.07 (*S. skyliuae* and *S. wugongshanensis*). All PCGs used ATN as the initiation codons, except for *nad1* (*S. raja* and *S. sinensis*), which begins with TTG, and *atp6* (*S. raja*), which begins with GTG. Only *atp6* and *cox2* (*S. skyliuae*) had incomplete stop codon with T residue; other PCGs terminated with TAA/TAG. The incomplete termination codons are presumed to be filled by polyadenylation during the mRNA maturation process [34].

The RSCU of the four Scolytoplatypodini species was calculated (Figure 2). The codons that were most commonly used were UUA-Leu, UUU-Phe, AUU-Ile, and AUA-Met. This result indicated that UUA is the most preferred codon. Additionally, there was a strong A/T bias in the PCGs.

### 3.3. Nucleotide Diversity (Pi) and Nonsynonymous (Ka)/Synonymous (Ks) Mutation Rate Ratios

The Pi of the four Scolytoplatypodini species based on 13 PCGs was computed (Figure 3), and it ranged from 0.15 to 0.32. Among the PCGs, *atp8* (0.32) had the highest Pi values, followed by *nad6* (0.26) and *nad2* (0.25). The *cox1* (0.15) had the lowest Pi values, which implies that *cox1* is the most conserved gene in *Scolytoplatypus*.

The ratios of Ka/Ks for every gene of the 13 PCGs were also computed (Figure 3). The values of Ka/Ks ranged from 0.05 to 0.52; 13 PCGs displayed low evolutionary rates (0 < ω < 1), indicating evolutions of 13 PCGs under the purification option [35]. *Cox1* (0.05) exhibits the lowest evolutionary rate, suggesting that it experienced the strongest purifying selection. The *atp8* (0.52) and *nad6* (0.38) exhibited a faster rate of evolution.

### 3.4. Gene Overlaps and Intergenic Spacers

Gene overlaps were found in all four mitogenomes, and every overlap ranged from 1 bp to 7 bp (*S. raja*, seven gene junctions, 27 bp overlaps; *S. sinensis*, seven gene junctions, 21 bp overlaps; *S. skyliuae*, seven gene junctions, 23 bp overlaps; *S. wugongshanensis*, 10 gene junctions, 26 bp overlaps;). All mitogenomes shared the same two types of gene overlaps: *atp8*-*atp6* (7 bp) and *nad4*-*nad4l* (7 bp). Gene overlaps were also found in other known Scolytinae mitogenomes [15,27,28,29].

Intergenic spacers were identified in the four mitogenomes, including 14 intergenic spacers in *S. raja*, 15 in *S. sinensis*, 17 in *S. skyliuae,* and 13 in *S. wugongshanensis*. The length of intergenic spacers ranged from 1 bp to 68 bp (Appendix A). The longest intergenic spacer was located between *trnS2* and *nad1* in *S. raja*.

### 3.5. Transfer RNA, Ribosomal RNA Genes, and Non-Coding Regions

The position and secondary structures of tRNA genes were identified by the MITOS server. The mitogenomes of the four species each contained 22 typical tRNAs, and 14 tRNAs were encoded by the J-strand; the others were encoded by the N-strand (Figure 1 and Appendix A). The length of the four Scolytoplatypodini mitogenomes ranged from 59 bp (*trnS1*, *trnY*) to 71 bp (*trnQ*) in *S. raja*, from 59 bp (*trnY*) to 71 bp (*trnQ*) in *S. sinensis*, from 55 bp (*trnS1*) to 68 bp (*trnQ*, *trnM*, *trnK*) in *S. skyliuae*, and from 57 bp (*trnS1*) to 70 bp (*trnK*) in *S. wugongshanensis* (Appendix A).

A typical tRNA consists of a discriminator nucleotide, amino-acyl (AA) arm, TΨC (T) arm, variable (V) arm, anticodon (AC) arm, and dihydrouridine (DHU) arm. All tRNAs exhibit a canonical cloverleaf structure, except for *trnS1* lacking the DHU arm, it is common in most metazoan mitogenomes for *trnS1* to lack the DHU arm [8,11,36]. Such abnormal tRNAs may sustain their function through a posttranscriptional RNA editing mechanism [37,38]. Nine mismatched base pairs (UU, GG, AA, UG, UC, AG, AC, two A, and single U) of tRNAs were found in the four Scolytoplatypodini mitogenomes. The *trnY* not only lacks the discriminator nucleotide but also lost three bases, so the three bases (GGU) of the 5’ end are exposed (Figure 4). The *trnA* (*S. raja*) and *trnK* (*S. sinensis* and *S. skyliuae*) also lack the discriminator nucleotide.

The *rrnL* genes of the four Scolytoplatypodini mitogenomes are located at the intergenic region between *trnL1* and *trnV,* with lengths that range from 1290 bp to 1313 bp. The *rrnS* genes were located between *trnV* and the D-loop, with sizes ranging from 753 bp to 760 bp. The two rRNAs are all encoded on the N-strand and have a high A/T bias that reached 81.5% in *S. raja*, 80.9% in *S. sinensis*, 80.3% in *S. skyliuae,* and 80.5% in *S. wugongshanensis*. Since rRNAs do not have functional annotation features like PCGs, it is so hard to establish their boundaries [11,39].

The D-loop acts in the initiation and regulation of replication and transcription in insects [12,40]. The D-loops of the four Scolytoplatypodini mitogenomes are all located between the *rrnS* and *trnI*. The full lengths of the D-loops are 586 bp in *S. raja*, 621 bp in *S. sinensis*, 495 bp in *S. skyliuae*, and 670 bp in *S. wugongshanensis.* The AT content ranged from 81.2% (*S. raja*) to 85.3% (*S. sinensis*). There were different lengths of repeat sequences in the D-loop.

### 3.6. Phylogenetic Analysis

The phylogeny of Scolytinae, based on the 13 PCGs, was constructed (Figure 5 and Appendix A) using 32 Scolytinae species and two out-groups (*Sitophilus oryzae* and *Sitophilus zeamais*). The ML tree has a similar topology to the BI tree, and their support values are reported above and below the nodes. Compared to the ML tree, the BI tree had higher confidence values, and the monophyly of most tribes and genera of the species studied was well supported. The node support values of BI trees were always higher than ML trees, especially in low-valued branches in the ML tree.

The clades consisting of Hylurgini, Hylastini, and Scolytoplatypodini had a high support value. However, within each of the close clades, Xyloterini and Cryphalini, the internal relationship among the included group was not strongly supported in the ML results. The monophyly of Scolytoplatypodini was confirmed. All *Scolytoplatypus* were clustered on one branch, and the nodes received high support. The *Scolytoplatypus* split into two branches; one branch was composed of *S. skyliuae* and *S. wugongshanensis,* and, despite their morphological similarities, the molecular data were diagnostic. The other branch was composed of *S. raja* and *S. sinensis*. The branch of *S. skyliuae* and *S. wugongshanensis* formed a sister lineage with *S. raja* and *S. sinensis.* These results support those of previous morphological and phylogenetic studies [1]. The Scolytini was the first diverging lineage in the subfamily Scolytinae lineage and was highly supported. This result was consistent with previous studies [41,42,43]. The Ipini, Dryocoetini, and Xyleborini were clustered in a branch and had high support values. This was consistent with the results of Du et al. [15]. The Corthylini and Polygraphini had low support values, although they were clustered together.

This is the first analysis of the relationship between Scolytoplatypodini in Scolytinae based on the mitogenome. However, the relationship between the Scolytoplatypodini and the other partial groups remains unclear. This is primarily due to the limited number of published mitogenomes within the Scolytinae. This issue could be resolved when the mitogenomes of additional Scolytinae species are obtained.

## 4. Conclusions

In this study, we sequenced and analyzed the mitogenomes of *S. raja*, *S. sinensis*, *S. skyliuae*, and *S. wugongshanensis*, in which mitogenomes had common and similar structures. Gene rearrangements are unusually rare in the mitogenomes in Coleoptera, especially PCGs. [44]. The complete mitogenomes of the four Scolytoplatypodini species, like other Scolytinae, revealed high A/T bias [15,45,46]. Most PCGs start with ATN except for *nad1* in *S. raja* and *S. sinensis,* which begin with TTG, and for *atp6* in *S. raja,* which begins with GTG. Only *atp6* and *cox2* (*S. skyliuae*) are terminated with an incomplete T residue, other PCGs terminated with TAA/TAG. By analysis of KA/KS, *cox1* was determined to be the most conserved gene. In contrast, *atp8* and *nad6* had relatively higher evolutionary rates that differed from several bark beetles [15]. Most tRNA molecules have a typical cloverleaf structure, but *trnS1* lacks the DHU and *trnA* (*S. raja*) and *trnK* (*S. sinensis* and *S. skyliuae*) lack the discriminator nucleotide. The *trnY* lacks not only the discriminator nucleotide but also the three bases. While the *cox1* gene is the most conserved, the *atp8* and *nad6* have relatively higher evolutionary rates, and this differs from other bark beetles [15]. The phylogenetic tree using 13PCGs showed the monophyly of Scolytoplatypodini. Compared to the results of Du et al. [15], our use of 13 PCGs to construct the ML tree yielded the same results as obtained with the BI tree. This implies that the phylogenetic tree that was constructed using 13 PCGs obtained more stable results.

On the issue of whether Scolytoplatypodini belongs to Scolytinae or Platypodinae, there is no doubt that Scolytoplatypodini belongs to Scolytinae. However, Scolytinae and Platypodinae have the same convergent evolution as resemblant wood-boring. Nevertheless, current molecular data show that Dryophthorinae and Platypodinae are indeed sister groups, refuting the close relationship between Scolytinae and Platypodinae [7,14,47]. *Remansus* was the second genus in Scolytoplatypodini. Although no samples of *Remansus* were obtained in this study, Pistone et al. [41] showed *Scolytoplatypus* as a sister group to *Remansus* with high node support, then both clustering with other Scolytinae, for example, genera *Scolytodes* and *Gymnochilus*. Jordal [6] suggested that the African *Scolytoplatypus* and the Asian *Scolytoplatypus* might represent two unique genera. In males, the Asian *Scolytoplatypus* differ from the African *Scolytoplatypus* by having a strongly modified prosternum with nodules or hooked projections, by the longer and more triangular antennal club, and in all but two species by the large fovea on the anterolateral angle of the male pronotum [6]. African *Scolytoplatypus* and Asian *Scolytoplatypus* have previously been divided into two major phylogenetic branches [1,6]. However, few representative specimens were sequenced in this study, making it difficult to provide a more detailed discussion. The four new mitogenomes were from species collected in Asia. It is necessary to sequence more *Scolytoplatypus* species to explore the relationship of African *Scolytoplatypus* with the Asian *Scolytoplatypus*. 

## Figures and Tables

**Figure 1 genes-14-00162-f001:**
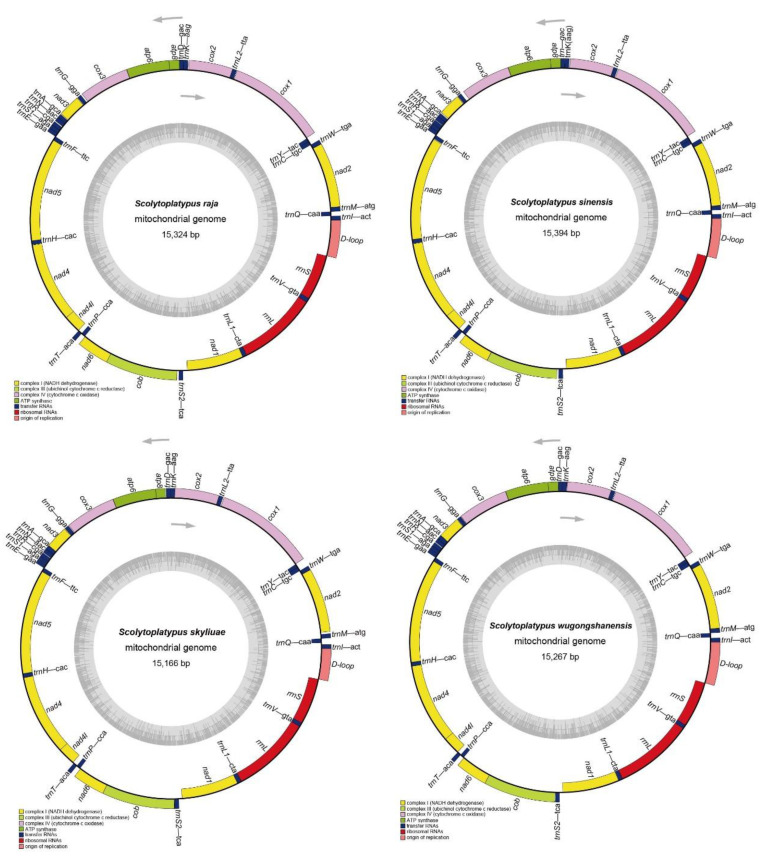
Circular map of the complete mitogenome of *S. raja*, *S. sinensis*, *S. skyliuae, and S. wugongshanensis*. Different colors indicate different types of genes and regions. Genes in the outer circle are located on the J-strand, and genes in the inner circle are located on the N-strand.

**Figure 2 genes-14-00162-f002:**
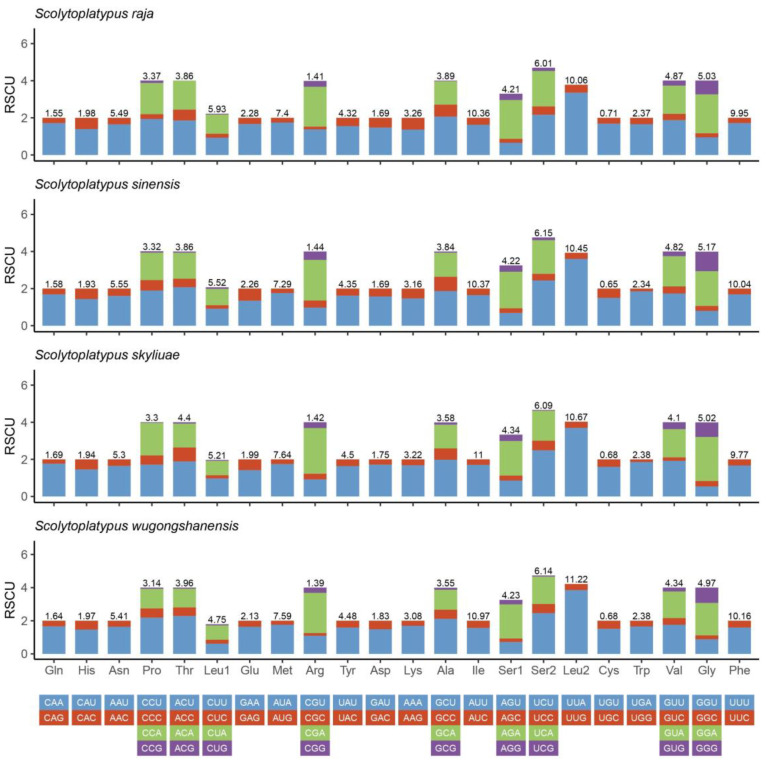
Relative synonymous codon usage (RSCU) of four *Scolytoplatypus* species. The ordinate represents the RSCU (the number of times a certain synonymous codon is used/the average number of times that all codons encoding the amino acid are used). The abscissa represents different amino acids. The number above the bar graph represents the ratio of amino acids (number of certain amino acids/total number of all amino acids).

**Figure 3 genes-14-00162-f003:**
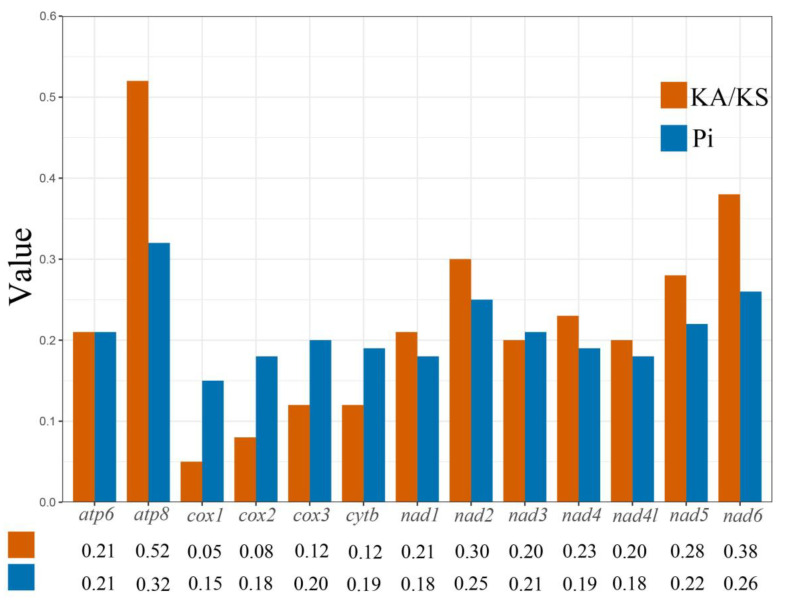
Nucleotide diversity (Pi) and nonsynonymous (Ka)/synonymous (Ks) mutation rate ratios of 13PCGs of *Scolytoplatypus* species (the Pi and Ka/Ks values of each PCG are shown under the gene name).

**Figure 4 genes-14-00162-f004:**
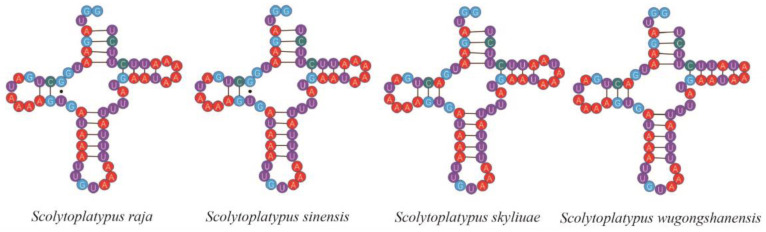
Secondary structures of the *trnY* in *Scolytoplatypus* mitogenomes.

**Figure 5 genes-14-00162-f005:**
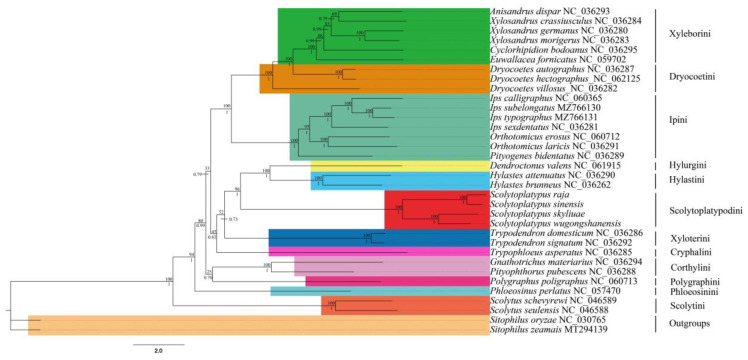
Phylogenetic tree of Scolytinae inferred from ML and BI methods based on 13 PCGs. The ML tree has the same topology as the BI tree, and their support values are reported above and below the nodes, respectively.

**Table 1 genes-14-00162-t001:** Voucher information of the specimens used for mitochondrial genomes sequencing.

Specimens	Location	Geographic Info.	Date of Collection
*S. raja*	Gaoligong National Nature Reserve, Yunnan	25.29 N, 98.80 E	27 July 2019
*S. sinensis*	Leibo County, Sichuan	28.41 N, 103.77 E	5 August 2021
*S. skyliuae*	Wuyishan National Nature Reserve, Jiangxi	27.88 N, 117.78 E	17 July 2017
*S. wugongshanensis*	Wugong Mountain, Jiangxi	27.58 N, 114.23 E	27 September 2017

**Table 2 genes-14-00162-t002:** Mitochondrial genome information used in this study.

Subfamily	Species	Length (bp)	GenBank Accession No.	References
Scolytinae	*Anisandrus dispar*	16,665	NC_036293	Unpublished
Scolytinae	*Xylosandrus crassiusculus*	16,875	NC_036284	Unpublished
Scolytinae	*Xylosandrus germanus*	16,099	NC_036280	Unpublished
Scolytinae	*Xylosandrus morigerus*	16,246	NC_036283	Unpublished
Scolytinae	*Cyclorhipidion bodoanus*	15,899	NC_036295	Unpublished
Scolytinae	*Euwallacea fornicates*	15,745	NC_059702	[27]
Scolytinae	*Dryocoetes autographus*	17,055	NC_036287	Unpublished
Scolytinae	*Dryocoetes hectographus*	16,040	NC_062125	[15]
Scolytinae	*Dryocoetes villosus*	15,859	NC_036282	Unpublished
Scolytinae	*Ips calligraphus*	19,144	NC_060365	[28]
Scolytinae	*Ips subelongatus*	16,040	MZ766130	[15]
Scolytinae	*Ips typographus*	16,793	MZ766131	[15]
Scolytinae	*Ips sexdentatus*	18,579	NC_036281	Unpublished
Scolytinae	*Orthotomicus erosus*	16,753	NC_060712	[15]
Scolytinae	*Orthotomicus laricis*	18,887	NC_036291	Unpublished
Scolytinae	*Pityogenes bidentatus*	18,781	NC_036289	Unpublished
Scolytinae	*Dendroctonus valens*	16,547	NC_061915	Unpublished
Scolytinae	*Hylastes attenuates*	17,409	NC_036290	Unpublished
Scolytinae	*Hylastes brunneus*	15,774	NC_036262	Unpublished
Scolytinae	*Scolytoplatypus raja*	15,324	OP719285	This study
Scolytinae	*Scolytoplatypus skyliuae*	15,166	OP719283	This study
Scolytinae	*Scolytoplatypus sinensis*	15,394	OP719284	This study
Scolytinae	*Scolytoplatypus wugongshanensis*	15,267	OP712675	This study
Scolytinae	*Trypodendron domesticum*	16,986	NC_036286	Unpublished
Scolytinae	*Trypodendron signatum*	16,909	NC_036292	Unpublished
Scolytinae	*Trypophloeus asperatus*	17,039	NC_036285	Unpublished
Scolytinae	*Gnathotrichus materiarius*	16,871	NC_036294	Unpublished
Scolytinae	*Pityophthorus pubescens*	17,316	NC_036288	Unpublished
Scolytinae	*Polygraphus poligraphus*	17,434	NC_060713	[15]
Scolytinae	*Phloeosinus perlatus*	17,054	NC_057470	Unpublished
Scolytinae	*Scolytus schevyrewi*	15,891	NC_046589	[29]
Scolytinae	*Scolytus seulensis*	16,396	NC_046588	Unpublished
Dryophthorinae	*Sitophilus oryzae*	17,602	NC_030765	[30]
Dryophthorinae	*Sitophilus zeamais*	18,531	MT294139	[30]

## Data Availability

The mitogenomes were deposited at NCBI, with accession numbers OP719285, OP719283, OP719284, and OP712675.

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
