# Peer review of "Complete Mitochondrial Genome of Scolytoplatypodini Species (Coleoptera: Curculionidae: Scolytinae) and Phylogenetic Implications"

_genes, 2023, doi:10.3390/genes14010162_

Round 1
Reviewer 1 Report
This study represents the collection of the first mitogenome sequences of this particular genus, to which research has been fairly limited. While it won't redefine or resolve its taxonomy classification (for that, as well said in the Conclusion, more sequences are needed), it could be an important source for future studies. Still, the genetic characterization of the mitogenome is well done.
My most important aspect to revise is the actual taxonomy classification. It's still not clear where this particular genus falls into, as I have not found a definitive study which analyses their phylogenetic assessment into the Curculionidae family. Some sources (like the Lifemap: https://lifemap-ncbi.univ-lyon1.fr/?tid=141182) place the Scolytoplatypus genus into the Platypodinae tride, unlike has been supported in the past (as the authors implied, it was part of the Scolytinae, which maybe is not the case anymore). On another note, the Remansus genus is indeed still classified as a genus in the Scolytinae family, which may indicate it is not so closely related to Scolyplatypus as previously though. While it did not directly impact the importance of the study (mitogenomes), I would just advise the authors to more throughly explore this and revise the text (Introduction and Conclusion) if needed.
Minor revisions:
35 to 37 - Make this phrase more clear, for example "Scolytoplatypodini species usually attack small stems, but only some species, which are secondary borers, seem to attack healthy living trees [5]."
73 - "...identified..."
265 - "This issue could be resolved..."
270 - "...had common and similar..."
285 - "This implies that the phylogenetic tree that was construced using 13 PCGs got more stable results."
Author Response
Response to Reviewer 1 Comments
Point 1: My most important aspect to revise is the actual taxonomy classification. It's still not clear where this particular genus falls into, as I have not found a definitive study which analyses their phylogenetic assessment into the Curculionidae family. Some sources (like the Lifemap: https://lifemap-ncbi.univ-lyon1.fr/?tid=141182) place the Scolytoplatypus genus into the Platypodinae tride, unlike has been supported in the past (as the authors implied, it was part of the Scolytinae, which maybe is not the case anymore). On another note, the Remansus genus is indeed still classified as a genus in the Scolytinae family, which may indicate it is not so closely related to Scolyplatypus as previously though. While it did not directly impact the importance of the study (mitogenomes), I would just advise the authors to more throughly explore this and revise the text (Introduction and Conclusion) if needed.
Response 1: Thank you very much for your constructive comment. About Scolytoplatypus belongs to Scolytinae or Platypodinae? I think it is nodoubt that Scolytoplatypus belongs to Scolytinae. First, despite Scolytinae and Platypodinae have same convergent evolution as resemblant wood-boring. Previous comparative analyses of morphological data focused to a large extent on adult head structures, features that are heavily modified through adaptation to wood boring and therefore not necessarily homologous in taxa with similar feeding behaviour. But previous molecular studies provided the clearest evidence to date for a sister relationship between Dryophthorinae and Platypodinae. Such as, Haran et al., (2013) and Gillett et al., (2014) used mitogenome to elucidate the Phylogeny of Weevils (Coleoptera: Curculionoidea) Mugu et al., (2018) using 10 molecular markers resolve the phylogenetic position of the Platypodinae. With maximum support in the various analyses presented here, it concluded that these two subfamilies (Dryophthorinae and Platypodinae) are indeed sister groups. Molecular data therefore refute a close relationship between Scolytinae and Platypodinae. Then, Pistone et al., (2017) showed molecular phylogeny of Scolytinae based on 18 molecular markers. In this study, Scolytoplatypus as sister group to Remansus Jordal with high node support, then both clustering with other Scolytinae, for example genera Scolytodes and Gymnochilus. Moreover, phylogenies of bark and ambrosia beetles revealing the origins of fungus farming and subsociality, Scolytoplatypodini (Scolytoplatypus) was nested within parts of Hexacolini (genera Scolytodes and Gymnochilus). So we consider that Scolytoplatypus as sister group to Remansus in Scolytoplatypodini (Scolytinae) is dependable. We have added relevant content to the manuscript, line 285-293. At original experimental design, we didn’t intend to explore this issue. But, your comment is very farseeing, there are indeed no using mitogenome to elucidate the Phylogeny of Scolytoplatypus and Platypodinae. We will do it in next time with more Platypodinae’s mitogenome.
Point 2: 35 to 37 - Make this phrase more clear, for example "Scolytoplatypodini species usually attack small stems, but only some species, which are secondary borers, seem to attack healthy living trees [5]."
Response 2: Thank you for your suggestion. We have reorganized the sentence, line 36-39.
Point 3: 73 - "...identified..."
Response 3: Thank you for your kind comment. We have revised it into identified, line 73.
Point 4: 265 - "This issue could be resolved..."
Response 4: Thank you for suggestion. We have revised them in the manuscript, line 263-264.
Point 5: 270 - "...had common and similar..."
Response 5: Thank you for suggestion. We have revised them in the manuscript, line 268.
Point 6: 285 - "This implies that the phylogenetic tree that was construced using 13 PCGs got more stable results."
Response 6: Thank you for suggestion. We have revised them in the manuscript, line 283-284.

Reviewer 2 Report
Yu et al provide the first complete mitogenome analysis, with the presentation of four new beetle mitogenomes and their analysis in a genomic context. This study is well done, and with some minor edits should be published. My comments below.
Major:
--Acknowledgements mention Chengpeng Long, who performed genome assembly. This person should be a co-author, not an acknowledgement, since the generation of the genome is the key first step. Who annotated the genomes? This should be listed in the acknowledgement and if it was Chengpeng, even more argument for making him a co-author.
--The author contributions section is very vague--please make it clearer who did what work.
--Given the lack of RNA-seq confirmation, the authors need to be very careful about over interpreting their annotations. For example, the gene overlaps may not be real; the presumed tRNA structures in Figure 4 could be different than expected. Authors should make clear these are hypothetical structures and not validated. This applies particularly to lines 213 - 217 where cloverleaf tRNA structures are discussed.
--Given the lack of experimental validation, gene overlaps may not be correct or the transcripts may terminate differently than the authors thing (i.e., at a 'T' rather than a full stop codon), please revise the overly-confident assertion of overlaps on lines 190 - 193. They may be true overlaps or they may not. Experimental work is needed to verify.
Minor:
--line 36…’small stems, most of which are secondary borers’ doesn’t make grammatical sense (seems to imply borders are the stems?)
--Phylogenetic analysis: use of the word ‘tribe’ on line 249 and following—seems strange. I believe the authors would cause less confusion by using the word ‘group’ instead. Tribe is idosyncratic; ulness there is a field-specific meaning I am not familiar with.
--Line 275-276, appears to claim both that termination is both with ’T’ alone and with TAA/TAG. Please clarify the sentence.
Author Response
Response to Reviewer 2 Comments
Point 1: --Acknowledgements mention Chengpeng Long, who performed genome assembly. This person should be a co-author, not an acknowledgement, since the generation of the genome is the key first step. Who annotated the genomes? This should be listed in the acknowledgement and if it was Chengpeng, even more argument for making him a co-author.
Response 1: Thank you for suggestion. We have added Chengpeng Long as an author.
Point 2: --The author contributions section is very vague--please make it clearer who did what work.
Response 2: Thank you for your suggestion. We have re-adjusted the author contributions
Point 3: Given the lack of RNA-seq confirmation, the authors need to be very careful about over interpreting their annotations. For example, the gene overlaps may not be real; the presumed tRNA structures in Figure 4 could be different than expected. Authors should make clear these are hypothetical structures and not validated. This applies particularly to lines 213 - 217 where cloverleaf tRNA structures are discussed.
Response 3: Thank you for your comment. The sequencing is accurate and the overlap is truly existing. The annotated results are very similar to other known Scolytinae mitochondrial genomes. We have annotated in the manuscript that the secondary structures of these tRNAs were derived from the predictions of MITOS, line 199-200, 207-209.
Point 4: --Given the lack of experimental validation, gene overlaps may not be correct or the transcripts may terminate differently than the authors thing (i.e., at a 'T' rather than a full stop codon), please revise the overly-confident assertion of overlaps on lines 190 - 193. They may be true overlaps or they may not. Experimental work is needed to verify.
Response 4: Thank you for your comment. We have adjusted the manuscript to make it look euphemistic, line 188-193. There were no issues with the sequencing quality of the rawdata. When we assembled the mitochondrial genome we used both MitoFlex and MitoZ assembly software to compare the assembly results. The assembled sequences were initially annotated using MITOS, followed by manual correction using Geneious. The gene overlaps phenomenon was present in all four mitochondrial genomes. The gene overlaps were also founded in all other known Scolytinae mitochondrial genomes, and professionals were consulted and concluded that gene overlap is common in mitochondrial genomes.
Point 5: --line 36…’small stems, most of which are secondary borers’ doesn’t make grammatical sense (seems to imply borders are the stems?)
Response 5: Thank you for your comment. We have reorganized the sentence, line 36-39.
Point 6: --Phylogenetic analysis: use of the word ‘tribe’ on line 249 and following—seems strange. I believe the authors would cause less confusion by using the word ‘group’ instead. Tribe is idosyncratic; ulness there is a field-specific meaning I am not familiar with.
Response 6: Thank you for suggestion. We have revised them in the manuscipt, line 247, 262.
Point 7: --Line 275-276, appears to claim both that termination is both with ’T’ alone and with TAA/TAG. Please clarify the sentence.
Response 6: Thank you for your comment. We have reorganized the sentence, line 273-274.
